# Behavior of Regular Insulin in a Parenteral Nutrition Admixture: Validation of an LC/MS-MS Assay and the In Vitro Evaluation of Insulin Glycation

**DOI:** 10.3390/pharmaceutics14051081

**Published:** 2022-05-18

**Authors:** Heloise Henry, Jean-François Goossens, Mostafa Kouach, Damien Lannoy, David Seguy, Thierry Dine, Pascal Odou, Catherine Foulon

**Affiliations:** 1Univ. Lille, CHU Lille, ULR 7365-GRITA-Groupe de Recherche sur les formes Injectables et les Technologies Associées, F-59000 Lille, France; jean-francois.goossens@univ-lille.fr (J.-F.G.); mostafa.kouach@univ-lille.fr (M.K.); damien.lannoy@univ-lille.fr (D.L.); thierry.dine@univ-lille.fr (T.D.); pascal.odou@univ-lille.fr (P.O.); catherine.foulon@univ-lille.fr (C.F.); 2Univ. Lille, Inserm, CHU Lille, U1286-INFINITE-Institute for Translational Research in Inflammation, F-59000 Lille, France; david.seguy@univ-lille.fr

**Keywords:** regular insulin, drug stability study, parenteral nutrition admixture, glycation, LC-MS/MS

## Abstract

Parenteral-nutrition (PN)-induced hyperglycemia increases morbidity and mortality and must be treated with insulin. Unfortunately, the addition of insulin to a ternary PN admixture leads to a rapid decrease in insulin content. Our study’s objective was to determine the mechanistic basis of insulin’s disappearance. The literature data suggested the presence of a glycation reaction; we therefore validated an LC-MS/MS assay for insulin and glycated insulin. In a 24-h stability study, 20 IU/L of insulin was added to a binary PN admixture at pH 3.6 or 6.3. When the samples were diluted before analysis with a near-neutral diluent, insulin was fully stable at pH 3.6, while a loss of around 50% was observed at pH 6.3. Its disappearance was shown to be inversely correlated with the appearance of monoglycated insulin (probably a Schiff base adduct). Monoglycated insulin might also undergo a back-reaction to form insulin after acidic dilution. Furthermore, a second monoglycated insulin species appeared in the PN admixture after more than 24 h at high temperature (40 °C) and a high insulin concentration (1000 IU/L). It was stable at acidic pH and might be an Amadori product. The impact of insulin glycation under non-forced conditions on insulin’s bioactivity requires further investigation.

## 1. Introduction

Patients receiving parenteral nutrition (PN) often require concomitant insulin treatment because of the nutritional product’s high dextrose content [1]. Indeed, hyperglycemia is observed in 50% of patients receiving PN, and is known to be harmful in both nondiabetic and diabetic individuals [2,3,4]. Various insulin formulations and regimens can be administered [5]: subcutaneously administered long-acting insulin [6,7,8,9,10,11] (reportedly associated with a greater frequency of hypoglycemic events [11], particularly when PN infusion has to be suspended [9]) or intravenous short-acting insulin (either via a Y-site insulin infusion [8,12,13] or direct addition of regular insulin (R-insulin) or its rapid-acting analogs to the PN admixture [7,10,11,14,15,16,17,18,19,20]). The guidelines issued by the American Society for Parenteral and Enteral Nutrition (ASPEN) and the European Society for Clinical Nutrition and Metabolism (ESPEN) emphasize the need for stability and efficacy data prior to the co-infusion of drugs and PN [21,22,23]. We previously used an immunoelectrochemiluminescence (IECL) assay to study the stability of R-insulin in a ternary PN admixture [24]; the insulin concentration fell by 40% over 6 h, but then did not change significantly up to 24 h. Moreover, pH, dextrose content, and time significantly influence insulin content, as described previously by Abdel-Wahab et al., for bovine insulin in dextrose solutions [25]. Given that PN contains high amounts of dextrose, glycation via a Maillard reaction might be involved in the loss of insulin, as has been described by Fry et al. for amino acids [26]. This phenomenon has been extensively described in vitro and in vivo for proteins in general [27,28,29] and for insulin in particular [30,31,32,33,34,35]. Most of the latter studies characterized the glycation products under “forced” conditions (i.e., high-temperature incubation for a long period of time) and/or reducing conditions. Nevertheless, the reaction’s early time course has not been described in detail [25,36].

To determine the mechanistic basis of the loss of native insulin in PN admixtures, a specific, orthogonal analytical method is required; it must be able to distinguish between a chemical modification and a conformation change. Indeed, the results of our IECL experiments could also be explained by a change in insulin’s three-dimensional structure that prevents the assay’s antibody from binding. If, on the contrary, a chemical modification is involved, it is important to identify the species formed.

Today’s insulin assays are based variously on radio-immunoassay (RIA) [25,37], immunoassay [38], electrophoretic [39,40], and chromatographic techniques [41,42,43,44,45,46]. Liquid chromatography-mass spectrometry (LC-MS) [47,48,49,50,51] appears to be the most suitable method because it potentially enables both quantification and compound identification with high levels of sensitivity, selectivity, precision, and robustness.

To this end, we optimized and validated a liquid chromatography-tandem mass spectrometry (LC-MS/MS) assay for the simultaneous measurement of R-insulin and (semi-quantitatively) glycated R-insulin. We first used this assay to study R-insulin’s behavior over time in a binary PN admixture; insulin was stable in acidic media but quickly disappeared in neutral media, with the simultaneous formation of glycated R-insulin. In a second step, we found that two different glycated R-insulin molecules were formed under forced conditions. Lastly, we studied the glycated molecules’ stability over time in acidic and neutral media: one was stable but the other disappeared and reverted to native R-insulin.

## 2. Materials and Methods

### 2.1. Chemicals

United States Pharmacopeia reference human insulin (H-insulin) and bovine insulin chain B oxidized (chain B, (purity ≥ 80%) were purchased from Sigma-Aldrich (Saint Quentin Fallavier, France) as lyophilized powders. The medicinal product Umuline Rapide^®^ (Eli-Lilly, Suresnes, France) was obtained as a 100 IU/mL (3.5 g/L) biosynthetic R-insulin solution in water for infusion, containing hydrochloric acid, sodium hydroxide, glycerol, and metacresol as excipients. The medicinal product Levemir^®^ (a 14.2 g/L biosynthetic detemir insulin solution prepared in water for infusion) was obtained from Novo Nordisk (Chartres, France). This marketed product contains glycerol, metacresol, phenol, zinc acetate, di-sodium hydrogen phosphate dihydrate, sodium chloride, hydrochloric acid, and sodium hydroxide (for pH adjustment). Anhydrous dextrose was obtained from Inresa (Bartenheim, France). Bovine serum albumin (BSA) V-fraction was purchased as a lyophilisate (95% pure) from Euromedex (Souffelweyersheim, France). Acetonitrile (ACN), hydrochloric acid (38%) and formic acid (purity: 99%) were obtained from VWR (Val de Fontenay, France), and trifluoroacetic acid (TFA; 99%) was purchased from Sigma-Aldrich. Ultrapure 18 MΩ water was produced by a Milli-Q system Millipore (Saint-Quentin en Yvelines, France).

Olimel^®^ N7E (a ternary PN admixture; a 1.5 L bag with three compartments: (i) 35% dextrose solution, Ca^2+^ and HCl, (ii) an 11% amino-acid solution with electrolytes, and (iii) a 20% lipid emulsion) and Cernevit^®^ lyophilisate (a vitamin mixture reconstituted in 5 mL of isotonic saline serum (0.9% NaCl)) were purchased from Baxter (Meyzieux, France). Nutryelt^®^ (10 mL trace element concentrate for a solution for infusion) was obtained from Aguettant (Lyon, France).

### 2.2. In Vitro Glycation of R-Insulin

Glycated R-insulin was obtained by reacting R-insulin with dextrose under forced conditions. A 5% dextrose solution (prepared with anhydrous dextrose in ultrapure water, with a final pH of around 6.0) was spiked with Umuline Rapide^®^ to obtain final R-insulin concentrations of 35 mg/L at two dextrose concentrations (5% and 17.5%). The solutions were incubated at 40 °C for 2 h or 24 h.

### 2.3. LC-MS/MS Apparatus and Method

The LC-MS/MS analyses were performed on a UFLC-XR Prominence^®^ system (Shimadzu, Kyoto, Japan) coupled to a QTRAP^®^ 5500 MS/MS hybrid system triple quadrupole/linear ion trap mass spectrometer (AB Sciex, Foster City, CA, USA) equipped with a Turbo VTM ion source operating in positive mode. Analyst software (version 1.5.2, AB Sciex) was used for system control, data acquisition, and data processing.

Compounds were separated using a C18 guard cartridge and a reversed-phase Kinetex C18 Core-shell column (100 × 2.1 mm i.d., 5 µm, 100 Å) (both from Phenomenex, Le Pecq, France). Gradient separation mode was used to separate native insulin (H- or R-insulin) from glycation products, using a 0.1% formic acid solution in ultrapure water (solvent A) and acetonitrile with 0.1% formic acid (solvent B). The elution gradient (flow rate: 0.4 mL/min) was as follows: (i) 0–3 min: 20% of solvent B; (ii) 3–13 min: a linear increase up to 80% of solvent B; (iii) 13–14 min: 80% of solvent B; (iv) 14–15 min: a linear decrease to 20% of solvent B; and (v) 15–16 min: 20% of solvent B. The autosampler and the column were thermostated at 5 °C and 25 °C, respectively. The injection volume was 10 µL.

Air was used as the nebulizer and heater gas mixture, while nitrogen was used as the curtain gas and collision gas. Under optimal conditions, the ion source nebulizer (GS1), heater (GS2), and curtain gas pressures were 50, 50, and 25 psi, respectively. The ion spray needle voltage, the declustering potential, and the cell exit potential were set to 5500 V, 100 V, and 15 V, respectively. A heater gas temperature of 450 °C was selected. H-insulin, glycated R-insulin, and the internal standard (IS: bovine chain B) were detected in multiple reaction monitoring (MRM) mode. The MRM transitions from the precursor to the dominant product ion, the type of transition (qualifying or quantifying), and the collision energies are summarized in Table 1. The experiment was repeated with R-insulin, in order to assess the equivalence of the transitions for H- vs. R-insulin.

### 2.4. Preparation of Calibration Standards (CSs) and Validation Standards (VSs)

#### 2.4.1. Preparation of the Binary PN Admixture

A binary (2-in-1) PN admixture was prepared by mixing equivalent volumes of two compartments of the Olimel^®^ N7E bag, i.e., (i) 35% dextrose solution, including Ca^2+^ ions, and (ii) a 11% amino acids solution, including electrolytes. The binary admixture was then spiked with the quantities of Cernevit^®^ and Nutryelt^®^ solutions commonly used in care units (i.e., 5 mL of Cernevit^®^ and 10 mL of Nutryelt^®^ for 1500 mL of PN admixture).

#### 2.4.2. Preparation of Standard Solutions

Standard solutions of the medicinal product Umuline Rapide^®^ R-insulin (3500 mg/L) and the IS chain B (2000 mg/L) were prepared in an ACN/H_2_O-20/80 (*v*/*v*) mixture with 0.1% formic acid. The CSs were prepared in a 10-fold diluted binary nutrition admixture (dilution solvent: ACN/H_2_O-20/80 (*v*/*v*) mixture with 0.1% formic acid and 0.3% BSA) and contained variously 0.010, 0.015, 0.025, 0.040, 0.060, 0.080, and 0.10 mg/L of R-insulin and 0.060 mg/L of IS. The VSs (prepared in the same medium) contained 0.010, 0.020, 0.070, and 0.10 mg/L R-insulin and 0.060 mg/L IS.

### 2.5. Validation of a Method for Assaying R-Insulin and Identifying Glycated R-Insulin Species

#### 2.5.1. The R-Insulin Assay

The R-insulin assay for the quantification of R-insulin and the semi-quantification of glycated insulin was validated in accordance with guidelines issued by the French Society of Pharmaceutical Sciences and Technologies (Société Française des Sciences et Techniques Pharmaceutiques, Paris, France) [52,53,54]. The validation was carried out on three consecutive days (in order to estimate prediction errors) with the H-insulin standard (to exclude the excipients contained in the R-insulin preparation) and then with R-insulin. Each day, seven CSs (each prepared in triplicate), four VSs (each prepared in triplicate), a matrix blank sample (binary PN admixture, 10-fold diluted in ACN/H_2_O-20/80 (*v*/*v*) mixture with 0.1% formic acid and 0.3% BSA; CAL00), and the same matrix blank sample spiked with IS (0.060 mg/L; CAL0) or with a mixture of R-insulin and glycated R-insulin at a total concentration of 0.016 mg/L (CAL1) were prepared and analyzed in MRM mode, by following the characteristic transitions of R-insulin (968.8 → 136.0 amu) and IS (1166.0 → 315.2 amu). Lastly, we studied the method’s specificity, response function, linearity, trueness, and precision (repeatability and intermediate precision). The acceptance criteria for precision and trueness were in line with European Medicines Agency’s guidelines [55]: a relative standard deviation (RSD) and a relative bias below 15% for the VS samples and below 20% for the lower limit of quantification (LLOQ). Lastly, accuracy profiles were assessed using NeoLiCy^®^ software (version 1.8.2.2, Spectra’min, Dijon, France), with acceptance limits of ±15% and a beta risk of 5%. The profiles were used to determine the LLOQ.

#### 2.5.2. Identification of Glycated R-Insulin Species

We also sought to validate the method for the detection and semi-quantification of glycated R-insulin. Hence, when validating the assay for R-insulin quantification, we monitored the glycated R-insulin transition (995.8 → 995.8 amu) in MRM mode in CAL0 and CAL1 solutions and a CS at the LLOQ of R-insulin.

### 2.6. Study of the Time Course of R-Insulin Glycation in a Binary PN Admixture at Two pHs

R-insulin glycation was studied in two different media. Medium 1 (a binary PN admixture at a non-acidic pH) was prepared from the first compartment of the Olimel^®^ N7E bag (containing a 35% glucose solution, Ca^2+^ ions, and HCl to set the pH to 3.6), mixed with an equivalent volume of the amino acid and electrolyte solution (leading to a pH 6.3 solution). Lastly, this mixture was spiked with Cernevit^®^ and Nutryelt^®^ solutions to give contents of 0.27% and 0.54%, respectively. An equivalent mixture (medium 2) at pH 3.6 (by the addition of 1 M HCl) was also prepared. In both media, the final concentration of glucose was 17.5%.

Each sample of medium was spiked with Umuline Rapide^®^ to give a final R-insulin concentration of 0.70 mg/L. For each mixture, a sample was collected immediately and analyzed to determine the concentration of R-insulin at t_0_. The solutions were then stored in a climatic chamber at 25 °C. Samples were collected and analyzed 1, 2, 4, 6, 18, and 24 h after insulin spiking. All assays were performed in triplicate.

Each sample was analyzed after one of two distinct dilution protocols: (i) 10-fold dilution in an acidic diluent (ACN/H_2_O-20/80 (*v*/*v*) mixture with 0.1% formic acid and 0.3% BSA (pH 2.70)) and (ii) 10-fold dilution in a neutral diluent (ACN/H_2_O-20/80 (*v*/*v*) mixture with 0.3% BSA (pH 7.35)). In each case, the final IS concentration was set to 0.060 mg/L.

The R-insulin concentration at the different contact times was determined from the peak area ratio (R-insulin/IS), using the previously established calibration curve. Lastly, we examined the R-insulin content expressed as a percentage of the initial concentration. We also measured the change over time in the glycated R-insulin/IS peak area ratio as a guide to the time course of glycation.

### 2.7. Study of the Stability of Glycated Insulin

Firstly, a 5% dextrose solution was spiked with Umuline Rapide^®^ to a final R-insulin concentration of 35 mg/L. After 24 h at 40 °C, two samples were collected. Each was 200-fold diluted in either an ACN/H_2_O-20/80 (*v*/*v*) mixture with 0.1% formic acid and 0.3% BSA (pH = 2.70) or an ACN/H_2_O-20/80 (*v*/*v*) mixture with 0.3% BSA (pH = 7.35). In each case, the final IS concentration was set to 0.060 mg/L. Each solution was then placed in the autosampler (set to 20 °C) for an analysis in MRM mode every 16 min for the first 1.5 h and then every 30 min until 6 h after the initial dilution, with quantification of R-insulin and semi-quantification of glycated R-insulin. The results were expressed as an area ratio A_analyte_/A_IS_.

## 3. Results

### 3.1. Development of An LC-MS/MS Assay for the Quantification of R-Insulin and the Identification of Glycated R-Insulin Species

The objective of the present study was to develop and validate an LC-MS/MS assay for R-insulin in a binary PN admixture, i.e., a method for determining the stability of H-insulin in solutions containing high dextrose concentrations and for simultaneously identifying the product formed.

#### 3.1.1. Optimization of the MS Detection Conditions

To optimize the detection parameters, solutions of H-insulin and the selected ISs (chain B and detemir, at a concentration of 7.0 mg/L in the ACN/H_2_O-20/80 (*v*/*v*) mixture with 0.1% formic acid) were first injected and analyzed in “full scan” positive mode. For the H-insulin, two main signals characteristic of [m + nH]^n+^ multi-charged ions were observed at *m*/*z* equal to 968.8 and 1162.5 amu; these corresponded to [M + 6H]^6+^ and [M + 5H]^5+^ ions, respectively, and were derived from H-insulin molecules (giving a mean molecular weight of 5807.2 Da). This finding is in line with the molecular mass of H-insulin given by the supplier (5807.7 Da). For chain B, three signals were obtained at *m*/*z* 700.1, 874.8, and 1166.1 amu, corresponding, respectively, to the multicharged-ions [M + 5H]^5+^, [M + 4H]^4+^, and [M + 3H]^3+^ (M_chain B_ = 3495.5 Da). For detemir, two signals were obtained at *m*/*z* 987.0 and 1183.9 amu for the [M + 6H]^6+^ and [M + 5H]^5+^ multicharged ions, respectively (M_detemir_ = 5916 Da).

For each observed multicharged ion, the optimal collision energies and MRM transitions from the precursor ion to the most abundant product ion (used for the quantification) were then determined in “product ion scan” mode. The values for H-insulin and chain B are shown in Table 1. For H-insulin, no fragmentation was observed for the multicharged precursor ion of *m*/*z* 1162.5 amu at any of the collision energies used. In contrast, the multicharged precursor ion of *m*/*z* 968.8 amu led to a main fragment with an *m*/*z* of 136.0 amu; this probably corresponded to a singly-charged immonium ion derived from the insulin chain B Tyr_26_. For chain B, the obtained fragment with *m*/*z* equal to 315.2 amu corresponded to the last three amino acids of the protonated (singly-charged) N-terminal chain B. For detemir, we selected the 1184.2 → 454.0 amu transition (collision energy = 44 eV).

For glycated R-insulin, the infusion mode was not suitable because the high dextrose concentration in the R-insulin-dextrose solutions (at least 1000 times the concentration of R-insulin) induces ion suppression. The MRM transitions had to be chosen in an LC-MS/MS experiment. This was performed under the optimal LC conditions described in the Materials and Methods section, with the injection of a 35 mg/L Umuline Rapide^®^ solution in 5% dextrose after a 2 h incubation at 40 °C. A “full scan“ experiment yielded two signals (*m*/*z* = 995.8 and 1194.6 amu), corresponding the [M + 6H]^6+^ and [M + 5H]^5+^ multicharged ions, respectively, that resulted from the ionization of a protein whose mean molecular mass was 5968.4 Da. In a Maillard reaction (Figure 1), a protein is glycated by the addition of a dextrose residue (M = 180 Da) and the loss of a water molecule (M = 18 Da), resulting in a molecular mass increase of 162 Da. The difference between the molecular masses of the formal products and initial R-insulin was 161.2 Da (i.e., 5968.4–5807.2). Hence, the molecule obtained by incubation of R-insulin in dextrose corresponds to monoglycated R-insulin.

#### 3.1.2. Optimization of the LC Assay

The chromatographic conditions were optimized in MRM mode, so that we could both quantify H- or R-insulin and detect/monitor glycated R-insulin with a short analysis time. Given that the molecules were detected in MRM mode, separation was not mandatory but was preferable (to avoid ion suppression phenomena). Hence, we sought operating conditions for the separation of H- or R-insulin and a potential IS. The stationary phase and the analytical conditions were selected with regard to the literature data on the LC-MS/MS analysis of H-insulin [49,51] and glycated (bovine or human) [30,32,33,35,59]. The separation was optimized on a Kinetex Core-shell C_18_, using various acidic ACN/H_2_O mixtures in gradient mode (injection of an equimolar mixture of H-insulin, chain B and detemir, each at 0.40 mg/L). An amount of 0.1% formic acid was preferred to 0.05% TFA because it gave a signal/noise ratio ten time higher. After optimization of the gradient program, the retention times of H-insulin, chain B, and detemir were, respectively, 5.59, 5.85, and 7.33 min. The resolution between H-insulin on one hand and chain B or detemir on the other was 2.25 and 24.5, respectively. Despite the better resolution for H-insulin and detemir, the asymmetry factor was higher for detemir (1.76) than for chain B (1.17). Furthermore, the peak intensity was 9-fold greater for chain B. Hence, chain B was selected as the IS.

Moreover, it is noteworthy that analyses of H-insulin and R-insulin gave the same chromatographic peaks (with the same retention times and peak areas); this showed that the additives present in Umuline Rapide^®^ (i.e., the excipients) did not perturb the analysis.

When monitoring the MRM transitions of glycated R-insulin under these chromatographic conditions, the chromatogram of 35 mg/L R-insulin incubated at 40 °C for 2 h in a 5% dextrose solution gave a peak at 5.86 min.

Lastly, a total runtime of 16 min was selected; this value enabled a return to the initial mobile phase composition and equilibration of the column.

Because we sought to develop a quantitative assay for R-insulin in binary PN admixtures, we also analyzed the latter matrix in “full scan” mode and under these optimal chromatographic conditions. As many ions were observed during the two first minutes of the analysis and, to avoid ion source fouling, the mobile phase eluted during this period was not introduced in the mass spectrometer, using a diverter valve.

#### 3.1.3. Optimization of the Diluent Composition for Standard and Sample Preparation

Given that the PN admixture has a high dextrose content and in order to avoid dextrose polymerization in the ion source (temperature: 450 °C), the samples had to be diluted 10-fold with mobile phase at the beginning of the separation procedure. A preliminary analysis of a solution containing 0.10 mg/L H-insulin and 0.060 mg/L IS in ACN/H_2_O-20/80 (*v*/*v*) mixture with 0.1% formic acid was performed immediately after preparation, and then 3 h later. The peak area ratio of H-insulin vs. IS (A_H-insulin_/A_IS_) was decreased by a factor of 2 after 3 h. In order to avoid H-insulin aggregation and adsorption of H-insulin onto the vial wall, 0.3% BSA was added to the samples; this protein was present in the samples used for the IECL H-insulin assay and was shown to be stable.

An equivalent solution (C_H-insulin-_ = 0.10 mg/L; C_IS_ = 0.060 mg/L) was prepared in the ACN/H_2_O-20/80 (*v*/*v*) mixture with 0.1% formic acid and 0.3% BSA. This solution was analyzed four times over a 3-h period. The coefficient of variation (CV) for the A_H-insulin_/A_IS_ ratio was 4.98%. Despite an increase in the noise is observed for retention times between 6.5 and 10.0 min (i.e., outside the elution range of H-insulin, IS, and glycated R-insulin), addition of BSA to the dilution solvent was not associated with ion suppression.

In conclusion, an ACN/H_2_O-20/80 (*v*/*v*) mixture with 0.1% formic acid and 0.3% BSA was selected for the preparation of standard solutions and the dilution of samples for analysis.

The chromatograms obtained under optimal conditions for a solution containing R-insulin, glycated R-insulin, and IS are shown in Figure 2.

### 3.2. Validation of the LC-MS/MS Method for the Quantification of R-Insulin and the Semi-Quantification of Glycated R-Insulin

Because the stability study was to be performed with Umuline Rapide^®^ (the medicinal products used in care units), it was important to take account of the preparation’s excipients. In preliminary experiments, we prepared standard calibration solutions with this commercially available product and the H-insulin standard. The two insulins gave equivalent results, and so the former was used to validate the assay.

#### 3.2.1. Quantification of R-Insulin

Firstly, the R-insulin assay’s specificity was assessed by comparing chromatograms obtained in MRM mode for R-insulin and IS, using a (i) matrix blank sample (binary PN admixture, 10-fold diluted in an ACN/H_2_O-20/80 (*v*/*v*) mixture with 0.1% formic acid and 0.3% BSA) (CAL00), (ii) the same matrix blank sample spiked with 0.060 mg/L IS (CAL0), and (iii) the same matrix blank sample spiked with R-insulin or glycated R-insulin at a total concentration of 0.16 mg/L (CAL1: C_R-insulin_ = 0.010 mg/L and C_glycated R-insulin_ = 0.0060 mg/L; the concentration of glycated-R-insulin was estimated from the mass balance) (Appendix A). For each transition monitored in MRM mode, the chromatogram for the matrix blank did not show a signal at the characteristic retention times for R-insulin and IS—thus demonstrating the specificity of the R-insulin assay.

In a second step, calibration curves for the peak area ratio (R-insulin/IS) vs. the R-insulin concentration were established using seven calibration standards prepared in the nutrient matrix and diluted 10-fold in an ACN/H_2_O-20/80 (*v*/*v*) mixture with 0.1% formic acid and 0.3% BSA. Our choice of the upper boundary of the calibration range (0.10 mg/L) was based on the mean R-insulin concentration used in the PN bags (0.70 mg/L) and the sample preparation procedure used in the H-insulin stability study (i.e., 10-fold dilution). The lower boundary of the calibration range (0.010 mg/L) corresponded to the disappearance of 85% of the R-insulin. Each solution was analyzed twice. The results obtained were modeled using the least squares method in a 1/X weighted linear model (Appendix A). All the R^2^ coefficients obtained during the three-day validation were greater than 0.987.

In a third step, we checked that the presence of glycated R-insulin did not influence the IS signal and thus the assay’s trueness. For example, co-elution of chain B and glycated R-insulin might have induced ion suppression phenomena. To this end, various amounts of glycated R-insulin were added to a 7 mg/L R-insulin solution. The peak area ratio (R-insulin/IS) increased linearly in proportion to the amount of glycated R-insulin added. This finding confirmed (i) the absence of ion suppression by glycated R-insulin and therefore (ii) the validity of the response function, whatever the quantity of glycated insulin in the sample.

The assay’s linearity (i.e., the method’s ability to give a quantitative results directly proportional to the true concentration of the analyte in the defined concentration range) was assessed by plotting the back-calculated concentration of the VS vs. the amount of VS introduced. An ANOVA including a correlation test (Fisher F1; *p* < 0.05), lack of fit tests (2nd order curve test and homogeneity of errors test (F2; *p* < 0.05)) was first performed. Secondly, using a Student’s test, a comparison of the straight line’s slope and intercept with the values of 1 and 0 (*p* < 0.05) attested to the assay’s linearity.

Trueness and precision were then determined at four concentration levels, from triplicate VS samples prepared each day. Trueness (expressed as the relative bias) was evaluated by comparing the nominal and back-calculated concentrations of R-insulin. Intraday and inter-day precisions were expressed as the RSD. As shown in Appendix A, the relative bias ranged from −6.01% to 0.48%, the intra-day precision ranged from 5.08% to 10.9%, and the inter-day precision ranged from 5.79% to 10.9%. Hence, the indices of trueness and precision were in line with the European Medicines Agency’s guidelines (<15%).

Lastly, the trueness and precision results were used to determine the accuracy profiles’ tolerance interval for a beta risk of 5%. The total error, which is the addition of both systematic and random errors, was calculated according to the SFSTP recommendations [52,53,54]. Whatever the concentration of the VS, the tolerance interval was included in the ±15% acceptance interval (Figure 3). Hence, the assay’s range for the quantification of R-insulin was 0.010–0.10 mg/L.

#### 3.2.2. Detection of Glycated R-Insulin

We also evaluated the assay’s specificity for the detection and semi-quantification of glycated R-insulin. Hence, during the validation of the R-insulin assay, the previously described CAL0 and CAL1 solutions and a CS (0.010 mg/L H-insulin) were analyzed in MRM mode by monitoring the glycated R-insulin pseudo-transition (995.8 → 995.8). The absence of a signal at glycated R-insulin’s retention time (5.86 min) in these solutions demonstrated the method’s specificity for glycated R-insulin.

### 3.3. The Time Course of R-Insulin Glycation in A PN Admixture

Firstly, R-insulin glycation was studied at 25 °C in a pH 6.3 binary PN admixture containing 17.5% dextrose, Ca^2+^ ions, HCl, amino acids, electrolytes, 0.27% Cernevit^®^, and 0.54% Nutryelt^®^ (medium 1). As soon as Umuline Rapide^®^ was spiked (0.70 mg/L) into this solution, several 100 µL samples were collected and analyzed after 10-fold dilution in either a pH 2.7 solution or a pH 7.3 solution. In each case, the final IS concentration was set to 0.060 mg/L. This strategy was based on two observations. Firstly, R-insulin was shown to be stable in acid medium [24]. The addition of acid was intended to stop the reaction responsible for R-insulin’s instability. Secondly, R-insulin’s instability has first been shown in our IECL study, the sample preparation protocol of which included dilution with a pH 7.5 phosphate buffer containing 4% BSA.

Figure 4 shows the change over time in R-insulin content (expressed as a percentage of the initial concentration). When the insulin solution was diluted in acidic media, the R-insulin content did not decrease significantly after 18 h. A 10% decrease was observed after 24 h of incubation. In contrast, dilution with a neutral diluent resulted in a 50% decrease in the R-insulin content after the first six hours (as also observed in our earlier IECL study) [24]. The R-insulin concentration remained constant between 6 and 24 h—as if a pseudo-equilibrium had been reached.

After dilution of the samples in non-acidic media, the chromatogram recorded for the glycated R-insulin transition (995.8 → 995.8) always featured the peak characteristic of glycated R-insulin (hereafter referred to as “Glyc-ins_a_”) at 5.86 min. The area ratio (Glyc-ins_a_/IS) for this peak changed over time (Figure 5). The Glyc-ins_a_ signal increased as the native R-insulin disappeared, and the value was 2.5 times higher after 24 h than it was after 2 h. These results clearly established a link between R-insulin glycation and the decrease in R-insulin content observed during the IECL-based stability study.

We also studied an equivalent binary PN admixture whose pH had been adjusted to 3.6 by the addition of 1 M HCl (medium 2); the native R-insulin concentration did not change over the time range considered, and glycated R-insulin was not detected. This result is in line with our data from the IECL-based time-course study.

### 3.4. Study of the Stability of Glycated R-Insulin

A 35 mg/L insulin solution in 5% dextrose (pH 6) was incubated for 24 h at 40 °C. A high starting concentration of insulin was chosen in order to form a greater quantity of glycated insulin. The resulting solution was then diluted with either an acidic diluent or a neutral diluent. The solutions obtained were then analyzed at regular time intervals over 6 h.

After acidic and neutral dilutions of the solution obtained at 24 h and their immediate analysis, two peaks were detected when monitoring the glycated R-insulin transition in MRM mode. In addition to a Glyc-ins_a_ peak (detected at 5.86 min, as previously), a second peak (for a species referred to hereafter as “Glyc-ins_b_”) was detected at 5.67 min. The appearance of this new glycated insulin molecule might have resulted from our use of forced glycation conditions (i.e., a high temperature and a high insulin concentration) to obtain more glycated forms.

Concerning the two molecules’ respective stability profiles (as shown in Figure 6), dilution of the samples with an acidic IS solution led to a rapid decrease in the Glyc-ins_a_/IS area ratio and a concomitant increase in the R-insulin/IS area ratio; this reflected an increase in insulin content. The Glyc-ins_b_/IS area ratio did not change during this time. These findings demonstrate that Glyc-ins_a_ (whether formed under mild conditions or forced conditions) is unstable in an acidic milieu. The reaction that forms Glyc-ins_a_ is reversible, with reversion to insulin. In contrast, Glyc-ins_b_ is stable and so is formed under forced conditions by an irreversible reaction.

When the solution was diluted with a neutral IS solution, the concentrations of R-insulin and glycated R-insulin did not change significantly.

## 4. Discussion

Insulin is frequently added to PN admixtures as a treatment for iatrogenic hyperglycemia. According to the ASPEN and ESPEN guidelines, stability and efficacy data are mandatory before a drug can be co-infused with a PN product. In our previous IECL-based study of the stability of insulin in ternary PN admixtures, we observed a rapid decrease in insulin content [24]. In view of dextrose’s ability to react with protein and thus form glycated products, we next sought to determine whether this reaction took place when insulin was added to a PN admixture containing a high glucose concentration. It is important to bear in mind that the disappearance of insulin observed with the IECL assay might reflect either (i) a conformational change in insulin that prevented antigen/antibody binding, or (ii) a chemical modification (e.g., glycation) and a failure of the specifically anti-insulin antibody to bind to glycated insulin.

A highly sensitive, specific insulin assay was therefore required. Furthermore, the experiments had to be carried out under conditions similar to those encountered in care units or at home, when a low concentration of insulin (20 IU/L, 0.7 mg/L) is typically added to the PN admixture. Moreover, the method also had to be able to identify glycated insulin molecules. As suggested by Soboleva et al. [60], an orthogonal method (i.e., LC-MS/MS) is required and might fulfill both objectives in a single run. We therefore repeated some of our IECL experiments with LC-MS/MS. It should be noted that, for simplicity, we used a binary PN admixture and not the ternary PN admixture used in the IECL study. In fact, in the earlier stability study at a pH value of 6.3, the insulin content decreased similarly in ternary vs. binary PN admixtures.

We first had to optimize and validate the assay. To the best of our knowledge, the simultaneous quantification of insulin and the detection or semi-quantification of glycated insulin have not previously been described.

Although insulin and glycated insulin have been separated on the preparative scale using reverse phase LC-UV [32,33,35,59,61] or boronate affinity chromatography [25,30], they were not quantified simultaneously. In most cases, only insulin was quantified; the quantity of glycated insulin was determined by the mass balance method [35,59]. Furthermore, mass spectrometry (i.e., matrix-assisted laser desorption ionization—time-of-flight (MALDI-TOF) and/or electrospray ionization (ESI) mass spectrometry), was used to characterize glycated insulin [32,33,59,62] and/or to identify glycation sites after endoproteinase digestion [33,35,58,61,62].

The method developed here combines chromatographic separation on a C18 column with ESI-MS/MS detection, giving an analysis time of 10 min. This is the first method capable of the simultaneous, specific quantification of R-insulin and the semi-quantification of monoglycated R-insulin in a binary PN admixture. Although the concentration of glycated insulin cannot be determined directly, our method reveals relative changes over time in the glycated insulin content in a binary PN admixture.

We first used our new assay to study the behavior of insulin in a binary PN admixture under the conditions encountered during real-life care (0.7 mg/L insulin in a 17.5% complex dextrose solution; pH 6.3; 25 °C). After dilution of the samples in a neutral diluent (similar to that used in our IECL study [63]), the specific insulin assay gave much the same insulin concentration-time profiles as in the IECL stability study [24]): a major decrease (by ~50%) in the R-insulin concentration after the first 6 h and then a plateau (as if a pseudo-equilibrium had been reached) between 6 and 24 h. Hence, the insulin concentration profile observed in the IECL study did not result from a conformation change in insulin. Over the same period of time (and from the beginning of the time-course study onwards), we detected a peak characteristic of monoglycated R-insulin (Glyc-ins_a_) at 5.86 min. The semi-quantitative LC-MS/MS method showed that the monoglycated insulin content increases as the native R-insulin disappears. These results clearly establish the link between R-insulin glycation and the decrease in R-insulin observed during the R-insulin stability study performed using IECL. Surprisingly, when the samples were diluted in acidic diluent (because the addition of acid was supposed to stop the glycation reaction [33,35,59,62]), neither the disappearance of insulin nor the formation of glycated insulin was observed. This finding (i) suggests that the monoglycated insulin formed in the binary PN admixture at pH 6.3 is not stable and may revert to native R-insulin and (ii) is also consistent with the fact that no glycation was observed at a pH of around 3.0 under the same conditions.

To confirm this hypothesis, we looked at whether glycated insulin was stable over time. First, a larger quantity of glycated insulin was prepared by incubating a 35 mg/L insulin solution in 5% dextrose pH 6 for 24 h at 40 ° C (i.e., under forced conditions). Surprisingly, and using the same MRM transition, we detected two monoglycated R-insulin species: Glyc-ins_a_ (as already detected at 5.86 min under mild conditions) and another more polar species (Glyc-ins_b_) at 5.67 min after 24 h. To the best of our knowledge, Glyc-ins_b_ has not previously been described; this observation emphasizes the value of our analytical method. We then evaluated the stability of each of the monoglycated insulin derivatives. After acidic dilution, Glyc-ins_a_ disappeared as the native insulin concentration increased, but the Glyc-in 2 content did not change. Hence, Glyc-ins_a_ appears to be formed by a reversible reaction. In contrast, the irreversible formation of Glyc-ins_b_ requires forced temperature conditions and a longer contact time.

In order to understand insulin’s behavior in binary PN admixtures, we referred to the literature data. Protein glycation is initiated by a spontaneous nucleophilic addition reaction between the free amino group of a protein (generally the N-terminal or a lysine side chain) and the carbonyl group of a reducing sugar (Figure 1) [29]. This reaction rapidly forms a reversible aldimine (a Schiff base), which rearranges over a period of days [64] to produce a ketoamine group (i.e., an Amadori product) under non-reducing conditions or a beta-hydroxy group under reducing conditions. Glycation reactions for insulin have already been described in vitro and in vivo. The nature of the glycated product depended on the experimental conditions; reducing conditions led more frequently and more rapidly to multiglycated insulin (because these conditions force the transition from a Schiff base to an Amadori product) [33,61,65], whereas a longer contact time under non-reducing conditions led mainly to monoglycated insulin [32,33]. Hence, the formation of monoglycated insulin under our non-reducing conditions is in line with the literature data.

Nevertheless, our results are surprising because (i) insulin disappeared quickly (but only with a stable level after 6 h and with formation of Glyc-ins_a_) and (ii) a second, more polar monoglycated insulin appeared at a higher insulin concentration and a higher temperature. This has not been observed in previous studies, most of which focused on the identification and characterization of glycated insulin molecules obtained under forced and/or reducing conditions. The rapid but only partial glycation of insulin (despite the presence of excess dextrose) has already been reported by Abdel-Wahab et al. [25] (for human insulin) and Préta et al. (for insulin aspart) [66]. The latter findings are in line with both the kinetic aspects of Maillard reactions (described by Van Boekel et al. [64] and the reversibility of the glycation reaction (leading to the formation of Glyc-ins_a_) in acidic solutions; Glyc-ins_a_ might correspond to the Schiff base, and Glyc-ins_b_ might correspond to the Amadori product [56,64].

There are two main explanations for why Glyc-ins_b_ has not been described previously. Firstly, most researchers initiated the in vitro glycation reaction at a high temperature or under reducing conditions and then stopped it by adding acetic acid [32,59,61,62,67]. Hence, the Schiff base present in the incubation mixture would have disappeared, and only the Amadori product or the beta-hydroxy amine would have been isolated. Secondly, the few studies performed under mild, non-reducing conditions for a short period of time (resulting in the formation of the Schiff base and Amadori product) were based on R-insulin assay techniques that would have been unable to quantify glycated insulin at the same time [25,36,57].

These assessments raise questions about the impact of insulin glycation in PN admixtures on insulin’s bioactivity in hyperglycemic patients. Some studies have already been performed outside the context of PN. For example, in-vitro-glycated insulin has been injected into animals and humans. It is important to note that the glycated insulin injected was obtained under forced temperature/time conditions or reducing conditions. Accordingly, these studies doubtlessly involved the Amadori product or the beta-hydroxy amine. A decrease in insulin bioactivity of around 30% [32,34,59] was observed. This might be due to a change in signal translocation following the receptor-glycated insulin interaction [34] or to glycated insulin’s failure to bind to the insulin receptor [68]. Indeed, glycation can lead to aggregation of oligomeric species or amyloid fibrillation and has been described for advanced glycation end products [29]. To the best of our knowledge, Abdel-Wahab et al.’s study [25] is the only one to have measured the bioactivity of glycated insulin formed under mild conditions and in the absence of a reducing agent. The researchers reported that glycated insulin (corresponding probably to the Schiff base) was less biologically active and so might contribute to glucose intolerance in people with diabetes.

Our present results encourage us to perform an in vivo study of the effect of Schiff base formation in PN admixture bags (i.e., under non-reducing conditions and at ambient temperature) on insulin’s bioactivity. Given that PN admixture bags are sometimes prepared up to a week in advance (for use by patients at home), longer contact times should also be investigated.

## 5. Conclusions

We developed an LC-MS/MS assay for the quantitative determination of R-insulin and the semi-quantitative determination of monoglycated R-insulin in a binary PN admixture. The addition of R-insulin to a PN admixture led to immediate glycation, accounting for the previously described insulin instability in PN. The glycation occurs only at a near-neutral pH and mainly during the first few hours of contact. It leads to the formation of a Schiff base that can revert to insulin and dextrose after dilution in an acidic medium. The present study is the first to have described insulin glycation in a PN preparation under non-reducing conditions and at ambient temperature. Incubation at 40 °C leads to an Amadori product that remains stable—even after acidic dilution. The maintenance or loss of insulin’s bioactivity after glycation under these conditions should now be investigated.

## Figures and Tables

**Figure 1 pharmaceutics-14-01081-f001:**
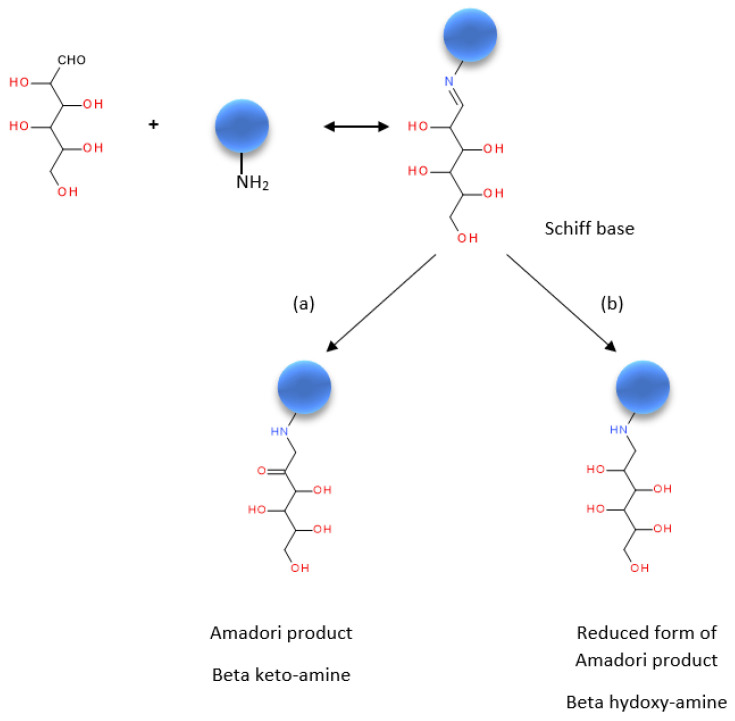
A Maillard reaction between a protein and dextrose [29,35,56,57,58] (**a**) under non-reducing conditions and (**b**) under reducing conditions. Blue spheres: protein.

**Figure 2 pharmaceutics-14-01081-f002:**
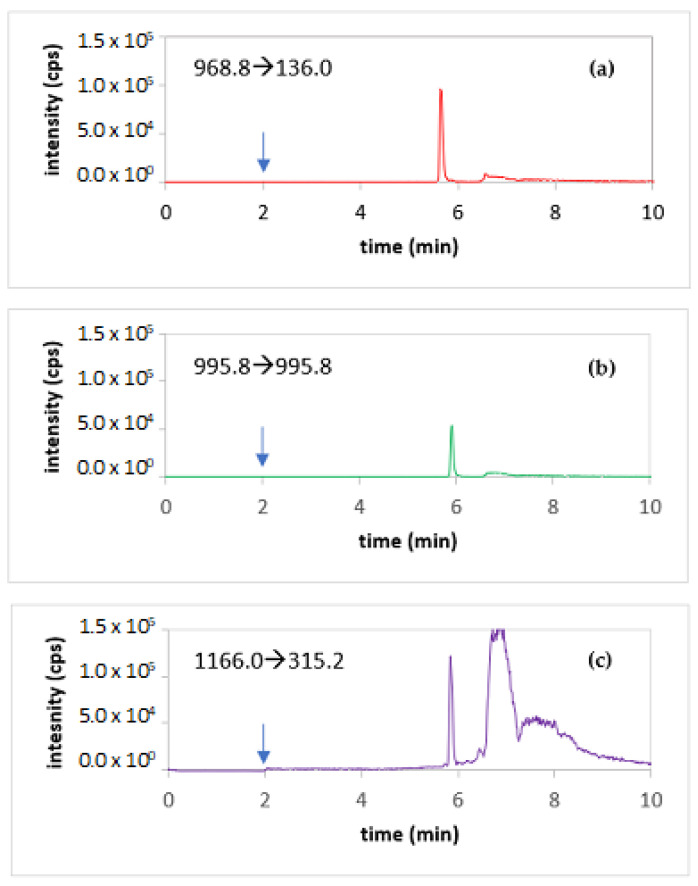
Chromatograms of a solution containing R-insulin, glycated R-insulin, and IS, with the MRM transition of (**a**) R-insulin (0.010 mg/L), (**b**) glycated R-insulin (0.0060 mg/L; concentration estimated from the mass balance), and (**c**) IS (0.060 mg/L) at t = 2 min: switching of the diverter valve to introduce of the mobile phase in the mass spectrometer (blue arrow).

**Figure 3 pharmaceutics-14-01081-f003:**
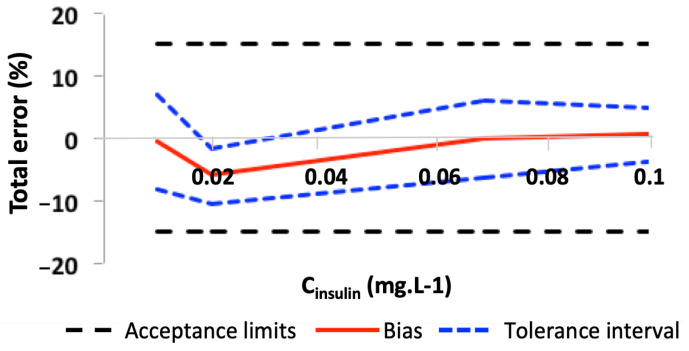
Accuracy profile for the LC-MS/MS R-insulin assay. The black dashed lines correspond to the ±15% acceptance limits, the red line corresponds to the bias, and the dashed blue lines correspond to the bias’s tolerance interval for a beta risk of 5%.

**Figure 4 pharmaceutics-14-01081-f004:**
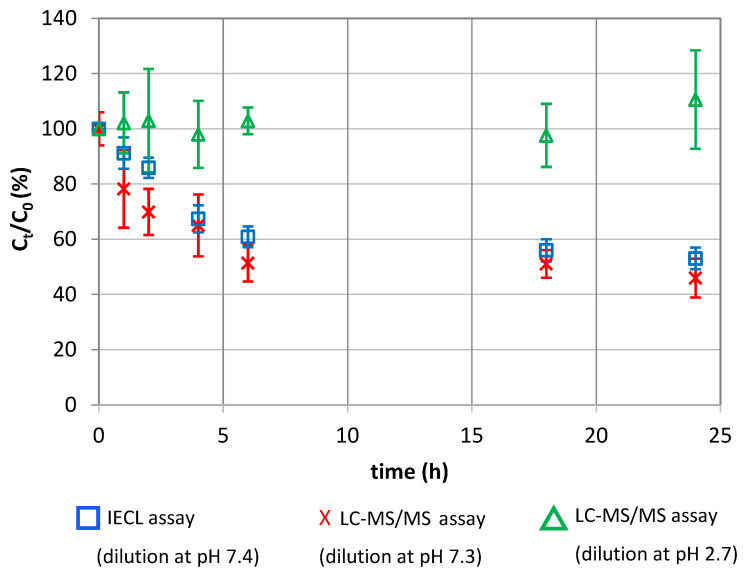
Stability of R-insulin (0.7 mg/L Umuline Rapide^®^ at t_0_) at 25 °C in a binary PN admixture (medium 1), containing 17.5% dextrose at pH 6.3; blue squares, data from the IECL assay after dilution in a pH 7.4 solution; red crosses, data from the LC-MS/MS assay after dilution in a pH 7.3 solution; green triangles, data from the LC-MS/MS assay after dilution in a pH 2.7 solution.

**Figure 5 pharmaceutics-14-01081-f005:**
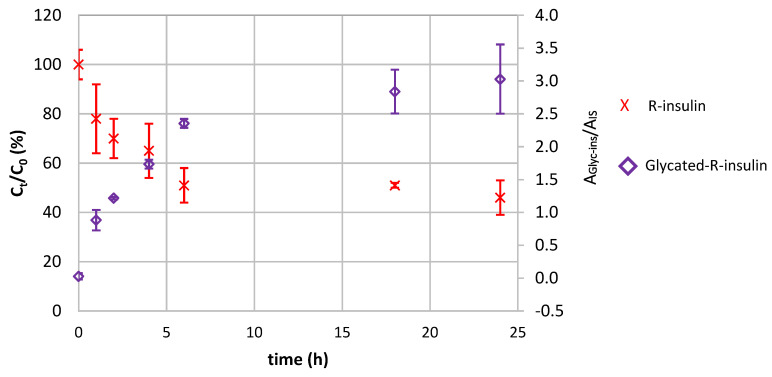
Change over time in the concentrations of R-insulin (red crosses) and glycated-R-insulin (“Glyc-ins_a_”, purple diamonds) in a binary PN admixture. Experimental conditions: 0.7 mg/L R-insulin at t_0_ in a binary PN admixture (medium 1) containing 17.5% dextrose at pH 6.3, and 25 °C; the LC-MS/MS analysis were performed after dilution with a pH 7.3 solution.

**Figure 6 pharmaceutics-14-01081-f006:**
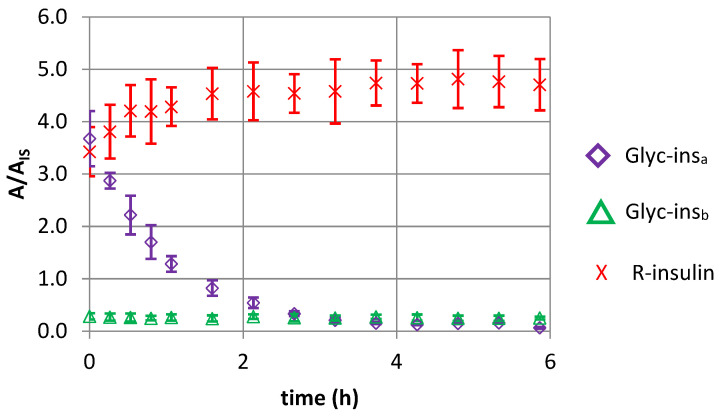
The change over time in the analyzed compound/internal standard area ratio (A/A_IS_) reflecting the concentration of glycated R-insulins (purple diamonds, Glyc-ins_a_, eluted at 5.67 min; green triangles, Glyc-ins_b_, eluted at 5.86 min) and R-insulin (red crosses) after acidic dilution of a solution of R-insulin in 5% dextrose 24 h after its preparation (incubation at 40 °C).

**Table 1 pharmaceutics-14-01081-t001:** Operating parameters, multiple reaction monitoring (MRM) transitions, and collision energies.

Molecule	Precursor Ion	Collision Energy (eV)	Transitions	
			Transition Type	*m*/*z* (amu)
H-insulin or R-insulin	[M + 6H]^6+^	39	Quantifying	968.8 → 136.0
	[M + 5H]^5+^	5	Qualifying	1162.5 → 1162.5
Glycated R-insulin	[M + 6H]^6+^	5	Quantifying	995.8 → 995.8
	[M + 5H]^5+^	5	Qualifying	1194.6 → 1194.6
Chain B	[M + 3H]^3+^	50	Quantifying	1166.0 → 315.2

H-insulin: human insulin (analytical standard); R-insulin: regular insulin (medicinal product); amu: atomic mass unit.

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
