# Peer review of "Behavior of Regular Insulin in a Parenteral Nutrition Admixture: Validation of an LC/MS-MS Assay and the In Vitro Evaluation of Insulin Glycation"

_pharmaceutics, 2022, doi:10.3390/pharmaceutics14051081_

Round 1

Reviewer 1 Report

The researchers present a method to quantify insulin and glycosylated insulin present in parenteral nutrition solutions, using liquid chromatography-mass spectrometry. The information reported regarding the validation of the method allows to see the novelty and relevance of the method.

I recommend publishing the manuscript without modifications.

Author Response

We thank you very much for your positive reviewing.

Reviewer 2 Report

Insulin loss in ternary PRN nutrition is an important problem, given that insulin is necessary to limit dextrose induced hyperglycemia in patients receiving PRN. 

Based on literature suggestion that insulin glycation might play a role in insulin stability in PRN,  the authors performed a painstaking analysis of insulin stability in low and mid pH as well as in PRN solutions as well. In order to perform these analyses, the authors first had to establish expected observation patterns for both insulin and glycated insulin.

The observed that insulin in unstable at PRN pH levels, whereas it is highly stable at low pH. As insulin levels dropped, glycated insulin levels increased in theirThey observed an inverse relationship between R-insulin levels and glycated R-insulin levels over time.

The authors of this manuscript performed an sound set of experiments that adequately supports their conclusions and sets the state for future work. I have no edits to recommend.

Author Response

(The authors gave the same response as above.)

Reviewer 3 Report

Please allow me to make the following minor comments to your submission:

Line 52: The sentence is incomplete, missing the predicate.

Line 100: The unit for resistance is inverted (MΩ).

Line 291: I am confused with the word "10-fold sensitivity". If this is referring to the signal output with the FA mobile phase in comparison to the TFA mobile phase, please consider reformulating the sentence. Also maybe the signal/noise ratio would be preferrable over raw signal in counts per second.

Line 363/Figure 2: Three traces are shown and zoomed to a common y-axis maximum which lets the reader compare the signal intensities quickly. But peak forms cannot be estimated from these graphs. Because of this I would recommend thinner lines. Also, the first two minutes show noise in purple and green traces but not in the red trace. In the Results part of the article the use of a diverter valve is implied. I would encourage the authors to explicitly state this and adjust Figure 2 accordingly.

Line 408: The half sentence in the brackets is incomplete.

Lines 411ff: The described comparison is not clear to me. Student's test is not the appropriate tool to test for assay linearity. Please elaborate on this.

Figure 3: Please explain the y-axis title below the figure or refer to the calculation that you elaborate in the paragraphs above it.

Figure 4/5: The y-axis is not uniformly labeled, once with redundant % units.

Line 487: The first part of the sentence does not add value to the information and could be omitted.

Figure 6: The x-axis is labaled with insignificant decimal places which is not uniform with the labels from former figures. Please decide to use one version in all of the figures. The label for the y-axis needs explaining somewhere.

Line 610: The "ii)" is missing a opening bracket.

Line 636: The adjective "doubtless" should be used as an adverb.

Line 663: I do not comprehend the meaning of this sentence. Should it be "... have described insulin glycation ..."?

Author Response

Dear Reviewer,

We wish to thank you for your report. We have taken your propositions into account. Please find the modifications that appear in red in the text of the attached file.

Kind regards.

The authors 

Reviewer 4 Report

p14, line 567: first six huors-> first 6 h

Author Response

Dear Reviewer,

We wish to thank you for your report. We have taken your proposition into account. Modification appears in red in the text.

  • p14, line 564: : first six hours-> first 6 h

As requested, we modified the sentence as follows: “A major decrease (by ~50%) in the R-insulin concentration after the first six hours 6 h and then a plateau (as if a pseudo-equilibrium had been reached) between 6 and 24 h.” (line 584)

Reviewer 5 Report

The only suggestion I have for the authors is to connect the dots to draw trend lines in several graphs in the manuscript. The error bars are present, but I would like to see the trend line clearly.

Author Response

Dear Reviewer,

We wish to thank you for your report. We prefer not to connect the dots as it will not help. Indeed, we prefer to avoid any ambiguity concerning the modelling of the kinetic profiles, which has not been carried out, and to make the results less legible.